# Immediate early gene *kakusei* potentially plays a role in the daily foraging of honey bees

**Asem Surindro Singh** [1,2]*, **Machathoibi Chanu Takhellambam**[3], **Pamela Cappelletti**[4], **Marco Feligioni**[4]

**1** National Centre for Biological Sciences, Tata Institute of Fundamental Research, Bangalore, India, **2** Department of Pathology, University of Mississippi Medical Center, Jackson, Mississippi, United States of America, **3** Department of Biotechnology, Manipur University, Canchipur, Imphal, India, **4** Laboratory of Neurobiology in Translational Medicine, Department of Neurorehabilitation Sciences, Casa Cura Policlinico, Milan, Italy

* asemsuren@gmail.com

**Data Availability Statement:** All relevant data are within the manuscript and its Supporting Information files.

**Funding:** This work was completed with the help of Research Associate Fellowship provided by Council

## Abstract

*kakusei* is a non-coding RNA that is overexpressed in foraging bee brain. This study describes a possible role of the IEG *kakusei* during the daily foraging of honey bees. *kakusei* was found to be transiently upregulated within two hours during rewarded foraging. Interestingly, during unrewarded foraging the gene was also found to be up-regulated, but immediately lowered when food was not rewarded. Moreover, the *kakusei* overexpression was diminished within a very short time when the time schedule of feeding was changed. This indicates the potential role of *kakusei* on the motivation of learned reward foraging. These results provide evidence for a dynamic role of *kakusei* during for aging of bees, and eventually its possible involvement in learning and memory. Thus the *kakusei* gene could be used as search tool in finding distinct molecular pathways that mediate diverse behavioral components of foraging.

## Introduction

Social behavior of honey bee foraging has been an attractive field of research. Further exploration of the dynamics of honey bee foraging at the molecular and cellular level could help in uncovering the complex mechanisms of social behavior. Honey bee foraging comprises several behavioral components including learning, memory, social interaction and communication. In 1973 the Austrian ethologist Karl Ritter von Frisch (Austrian ethologist) was honored with Nobel Prize in Physiology or Medicine for his investigations of sensory perceptions in honey bees [1]. He translated the meaning of the bee waggle dance into a particular movement of the honey bee that looks like the form of figure eight, and revealed that foraging bees of the same colony share information with the help of this dance [2–4]. Bee foraging has gained enormous attention, since it offers the opportunity to elucidate the complexity of behavioral mechanisms involved in accomplishing the foraging task. Thanks to the efforts of many researchers, today we have a substantial amount of knowledge on this subject. However, understanding of the context of molecular and cellular mechanisms underlying foraging tasks is still poor.

of Scientific and Industrial Research, Govt. of India, Award no. 09/860(0167)/2015—EMR-1 and Bridging Postdoctoral Fellowship by National Centre for Biological Sciences, TIFR, India, to Dr. Asem Surindro Singh.

**Competing interests:** The authors have declared that no competing interests exist.

During foraging, bees fly back and forth several times during the day between the hive and the food location, collecting nectar/pollen and bringing it to the hive for their colony. Foraging involves highly systematic and dynamic behavioral capacities that include long distance navigation using the sun as compass, evaluation of food quality, learning and memorizing flower cues, sense of colour, and social communication/interaction for coordination in collecting nectar, pollen, and water [2,5,6]. For decades the outdoor experiments have been routinely performed by feeding bees on pollen or sucrose solution placed in a specific place in order to mimic the natural foraging environment in close proximity. Using this experimental setup, we attempted to identify foraging regulatory genes in the European honey bee species *Apis mellifera* while the bees were performing their daily routine foraging. For this, the immediate early genes (IEGs) were the ultimate targets to examine, as they are well known as neural markers.

IEGs are rapidly induced by a large number of stimuli, and alterations of their expression are considered as part of the general neuronal response to natural stimuli as interpreted by normal synaptic activity [7]. The products of IEGs can activate downstream targets that typically function as part of a network of constitutively expressed proteins [7]. IEGs are also known to be the first activated genes that link cellular membrane events to the nucleus [8], and the gene expression changes are required for the late neuronal response which relates to the process of learning and memory formation [9] and which is a part of everyday brain functions [10]. In addition, depending on the type of the stimuli, the IEG-encoded proteins can be individually regulated in different parts of the brain [8,11], indicating that the same or different IEGs can, regulate different functions when expressed in different parts of the brain.

In our recent work, we have demonstrated involvement of two IEGs i.e., early growth response 1 (*egr-1*) and, nuclear hormone receptor 38 (*hr38*) and their corresponding partner genes such as ecdysone receptor (*ecr*), dopamine/ecdysteroid receptor (*dopecr*), dopamine decarboxylase (*ddc*) and dopamine receptor 2 (*dopr2*) during the daily foraging of bees [12]. Other papers have reported on the involvement of *egr-1* in the transition from nursing to foraging bees [13] and the role of *hr38* in caste and labor division [14], while the other genes were shown to have acted as *egr-1* downstream genes involved in ecdysteroid and dopamine signaling with a role in the processing of courtship memory [15] in Drosophila and in olfactory learning and memory [16] in honey bees, respectively. The IEG *c-jun* (also known as jun-related antigen, *jra* in fruit fly) and *egr-1* have been used as neuronal markers for the identification in honey bees of specific brain regions involved in the time memory as well as in innate and learned behaviours [17,18,19]. Interestingly, while most of the above mentioned genes are already well known through various research studies, a recently discovered insect IEG *kakusei*, a noncoding RNA, has been found to have function in the patterning of neuronal activity in the foraging bees [20,21,22]. Expression of *kakusei* was detected during ice-induced seizures in bees awakening from anesthesia, and its expression was prominent in the Kenyon cells of the mushroom bodies [20,21,22]. Since *kakusei* was found as neural marker, we were interested to extend the work on this gene in our experimental design.

The present study represents further investigation of immediate early genes (IEG) that could play a role during the daily foraging of honey bees, and is a continuous work of our previous report by Singh et al [12]. We selected two potential IEGs *kakusei* and *c-jun*, and four other orthologous genes that have been reported to be involved in cognitive process in vertebrates: extracellular signal-related kinase (*rk7*), glutamate receptor (*GluR*), 5-hydroxy tryptamine (serotonin) receptor 2 alpha (*5-HT2α*) and dopamine receptor 1 (*Dop1*). Among the six genes tested, we observed only for the *kakusei* gene with a transient and prolonged upregulation during reward foraging and a short period of overexpression during unrewarded foraging. These findings demonstrate a potential role for *kakusei* during reward foraging and in learning of food reward foraging.

## Materials and methods

### Foraging experiment and sample collection

Honey bees were purchased from the local bee keepers, Bangalore, Karnataka, India. The bee colonies were kept inside the bee house located within the institute campus, National Centre for Biological Sciences (NCBS), TIFR, Bangalore, India. And one of the most common honey bee species *Apis Mellifera* was used in this experiment. Therefore, no endangered or threatened species or locations were involved in this study. The behavioral tests were performed by using an outdoor flight cage (length = 12m, height = 5m, width = 2.5m) located within the NCBS campus, Bangalore, India. The *Apis mellifera* colonies were kept within the outdoor flight case and bees were fed with pollen and 1 M sugar solution placed on a green and yellow plastic plate respectively. The distance of feeders was 10m from the beehive with a 1.5m gap between the two feeders. The feeding time was from 14.00 to 17.00 hrs every day. Sample collection was started after the foraging bees had visited feeders for several days. For the gene expression profiling, nectar foragers were collected and the gene expression analysis was carried out with those samples. The procedures were same as previously described [12] and the same samples have been reused in this study.

### Collection procedure and sample grouping

**Collection during foraging.**   The first arriving foragers were caught at the feeder plate before presenting the sugar solution on the feeder (0min group) and the caught bees were immediately flash frozen in liquid nitrogen for further gene expression analysis. Soon after the first collection, sugar solution was presented and the first arriving foragers were marked by paint markers. The marked bees were collected at a series of time points with 15 min intervals up to 2 hrs (15min, 30min, 45min, 60min, 75min, 90min, 105min, and 120min groups), then flash frozen immediately in liquid nitrogen. We caught paint-marked foragers as soon as they landed over the plate and before they start drinking the sugar solution at those set times during their repeated trips. The paint mark on bees was done using Uni POSCA Paint Markers (Uni Mitsubishi Pencil, UK). In one day we collected about 24 bees, 1–2 bees for each time point starting from 14.00 hours until 16.00 hours and continued in following days until we obtained 5 bees at each time point.

**Collection before and after foraging.**   Pre-foraging bees were paint-marked while foraging and collected in the morning (09.00 hrs) of the next day in the hive before they started foraging; whereas post-foraging bees were caught in the hive in the evening (18.00 hrs) after the bees finished foraging, on the same day that the bees were paint marked. We took gentle care during the collection procedure in order that the foraging bees were not disturbed and to avoid inducing stress phenomena; thereby the interactions between the collector and the bees were considered minimal [23,24,25]. The caught bees were immediately flash frozen in liquid nitrogen.

**Collection without food reward.**   This category included only the foraging bees collected at the empty feeder plate at different time points. The foragers were marked at 0min upon their arrival at the empty feeder and 0min samples were collected. After the collection of 0min samples, 1 M sucrose solution was presented to let the bees continue coming and the collection continued for one hour with samples at 15min, 30min, 45min and 60min. While the unmarked bees were drinking, the marked bees were caught as soon as they landed on the feeder plate; it was made sure that those bees had not touched the sucrose solution. About 2 bees were collected in each day of collection and the collection started at 14.00 hrs. The bees were immediately flash frozen in liquid nitrogen as soon as they were caught.

**Collection at different feeding time.** Here the procedure is same with that of (1). The only difference was that the feeder was presented at 11.00 hrs instead of the normal feeding time 14.00 hours, and the collection also started from the same time and continued for 1 hr with samples at 0min, 15min, 30min, 45min and 60min.

## Brain dissection, RNA and cDNA preparation, and qPCR

The frozen bees were lyophilized at -50˚C with vacuum at 0.420 mBar for 20min, using a Freeze Zone® PlusTM 4.5 liter cascade Freeze Dry System (Labconco Corporation, Kanas City). Brain dissection was performed in a glass chamber containing 100% ethanol placed on a dry ice platform. The whole brain from each bee was dissected and placed into a micro centrifuge tube separately, and 500 μl Trizol (Trizol® Reagent, ambion RNA, life technology) was added. The brain was homogenized, RNA was extracted, and cDNA was prepared from that RNA, using the kits supplied by Invitrogen (Thermo Fisher Scientific) following the manufacturer's protocols. The cDNA from each brain was subjected to qPCR using a 7900HT Fast Real Time PCR System (Applied Biosystem, Singapore) in a 10μl reaction volume containing oligonucleotide primers (Sigma Aldrich) specific to target genes and SYBR Green (KAPA Syber® FAST PCR Master Mix (2X) ABI Prism®). The qPCR cycles followed Applied Biosystem protocol and the Rp49 gene was used as endogenous control [26] in each qPCR run and analysis. The details of the target genes and the oligonucleotide primers are provided in Table 1.

## Statistical analysis of qPCR

We calculated relative gene expression level using the relative standard curve method with the help of SDS 2.4 software provided with the 7900HT Fast Real system. The standard deviation was calculated following Applied Biosystem's 'Guide to performing relative quantification of gene expression using real-time quantitative PCR'. The fold change was determined relative to time t0, and the statistical significance was tested using one-way ANOVA with Turkey- Kramer post-hoc multiple comparison test; the analysis was carried out with the help of GraphPad InStat software [27]. Normal distribution of each group compared was tested using the D'Agostino & Pearson omnibus normality test. In order to check the differences among the

**Table 1. Details of primers used for quantifying the target genes.**

| Gene Name | NCBI Gene ID | Chromosome No. & location | Oligonucleotide primer sequence 5′ - 3′ | Amplicon size |
|---|---|---|---|---|
| *Kakusei* | 100049563 | LG2 | Fow—TGGGTAGGGTTGGTAAGGGAA | 91 bp |
| | | NC_007071.3 | Rev—ACACGAAACCATCCTGCCAC | |
| *Erk7* | 408917 | LG4 | Fow—ACCCGGTCCGAAGAAGAAAT | 67 bp |
| | | NC_007080.3 | Rev—CAGGCCAAAAGTCTGAGAATCA | |
| *c-Jun* | 726289 | LG9 | Fow—CCCTTCAGCAATTTAACCTTATC | 78 bp |
| | | NC_007078.3 | Rev—CGTGGCGGCATCCAAA | |
| *GluR* | 411220 | LG7 | Fow—GGGATCGCCTCATATACCCA | 71 bp |
| | | NC_007076.3 | Rev—GAGCGAACCAAAGGCTGTTT | |
| *5-HT$_2\alpha$* | 411323 | LG9 | Fow—GTCTCCAGCTCGATCACGGT | 126 bp |
| | | NC_007078.3 | Rev—GGGTATGTAGAAGGCGATCAGAGA | |
| *Rp49* | 406099 | LG11 | Fow—CAGTTGGCAACATATGACGAG | 124 bp |
| | | NC_007080.3 | Rev—AAAGAGAAACTGGCGTAAACC | |
| *Dop1* | 406111 | LG15 | Fow—ACAGAATTCCGAGAAGCGTTCA | 79 bp |
| | | NC_007084.3 | Rev–ATTCGCTAGTCGACGGTTCATTT | |

LG: Linkage Group

independent experiments, the two-way ANOVA program of GraphPad Prism was also employed (GraphPad Software Inc. www.graphpad.com).

## Results

In this study, only the immediate early gene *kakusei* showed a remarkable transient upregulation during the course of food reward foraging. The other five genes *erk7*, *c-jun*, *glur*, *5-ht2α*, *dop1* showed no statistically significant differences (S1 Fig).

### Expression profile of *kakusei* during food reward foraging

The expression of *kakusei* was measured at eleven time points, BF (pre/before foraging), 0min, 15min, 30min, 45min, 60min, 75min, 90min, 105min, 120min and AF (post/after foraging). In order to check the consistency of the results, three experiments were performed using independent samples collected from two different bee colonies over three months. The experiment 1 and 2 were from colony 1 and experiment 3 was from colony 2. Each of the three experiments demonstrated transient increases of *kakusei* level during the continuous food reward foraging over the collection time of two hours. The results are summarized in Fig 1 (S1 Table).

We observed an increase of *kakusei* expression from 0min (start of foraging, 14.00 hrs) to 15min (P = <0.0001) and from 15min to 30min (P = <0.0001) (Fig 1 Exp 1A). While the increase held at about 30min ($P_{30m-45m}$ = not significant) and a decrease was apparent from 45min, there was a sustained increased in expression of *kakusei* further down along the time series until 120min during the reward foraging as compared to the level at 0min (Fig 1 Exp 1A and S1 Table). A similar pattern was also observed in the subsequent two experiments with independent samples (Fig 1 Exp 2A/3A and S1 Table). The *kakusei* level was also significantly higher (P = <0.0001) at the start of foraging (0min) when compared to the level at pre foraging (09.00 hrs) or post foraging (18.00 hrs), while there was no significant difference between the before and after foraging samples (Fig 1 Exp 1B). This observation was further validated by two further independent experiments (Fig 1 Exp 2B & 3B). We then examined whether *kakusei* overexpression pattern was statistically different among the three independent results. We found significantly different overexpression of *kakusei* among the three independent experiments (P = 0.0001). The difference in *kakusei* levels between Exp-1/Exp-3 vs Exp-2 (P = 0.0001) was much higher than Exp-1 vs Exp-3 (P = 0.044). This difference might be due to age differences among the group of bees in Exp-1A/1B, Exp-2A/2B and Exp-3A/3B as we did not maintain the age of foraging bees during sample collection. The graphical results of the analysis are presented in Fig 2A. Similarly, in the case of the start/pre/post foraging groups, among the three independent experiments, we observed difference between Exp-2 vs Exp-1/-3 (P = 0.0001) and no difference between Exp-1 and Exp-3 (Fig 2B).

### Expression changes of *kakusei* during unrewarded foraging

In order to confirm whether the *kakusei* upregulation was solely due to the food reward, we presented the empty plate at the usual feeding time 14.00 hrs and the bees were collected without feeding following the procedure described in the methods section. Interestingly, *kakusei* expression from the whole bee brains showed upregulation at 15min (P = 0.0011; Fig 2C), indicating a potential effect on *kakusei* of the learned motivation of food reward. However, the value no longer increased after 15min ($P_{15m-30m}$/$P_{30m-45m}$/$P_{45m-60m}$ = not significant) and the level dropped by 45min ($P_{0m-45m}$ = not significant) (Fig 2C and S2 Table) indicating that a food reward is required in order to sustain *kakusei* upregulation. This result suggests that *kakusei* is involved in learning of food reward during foraging.

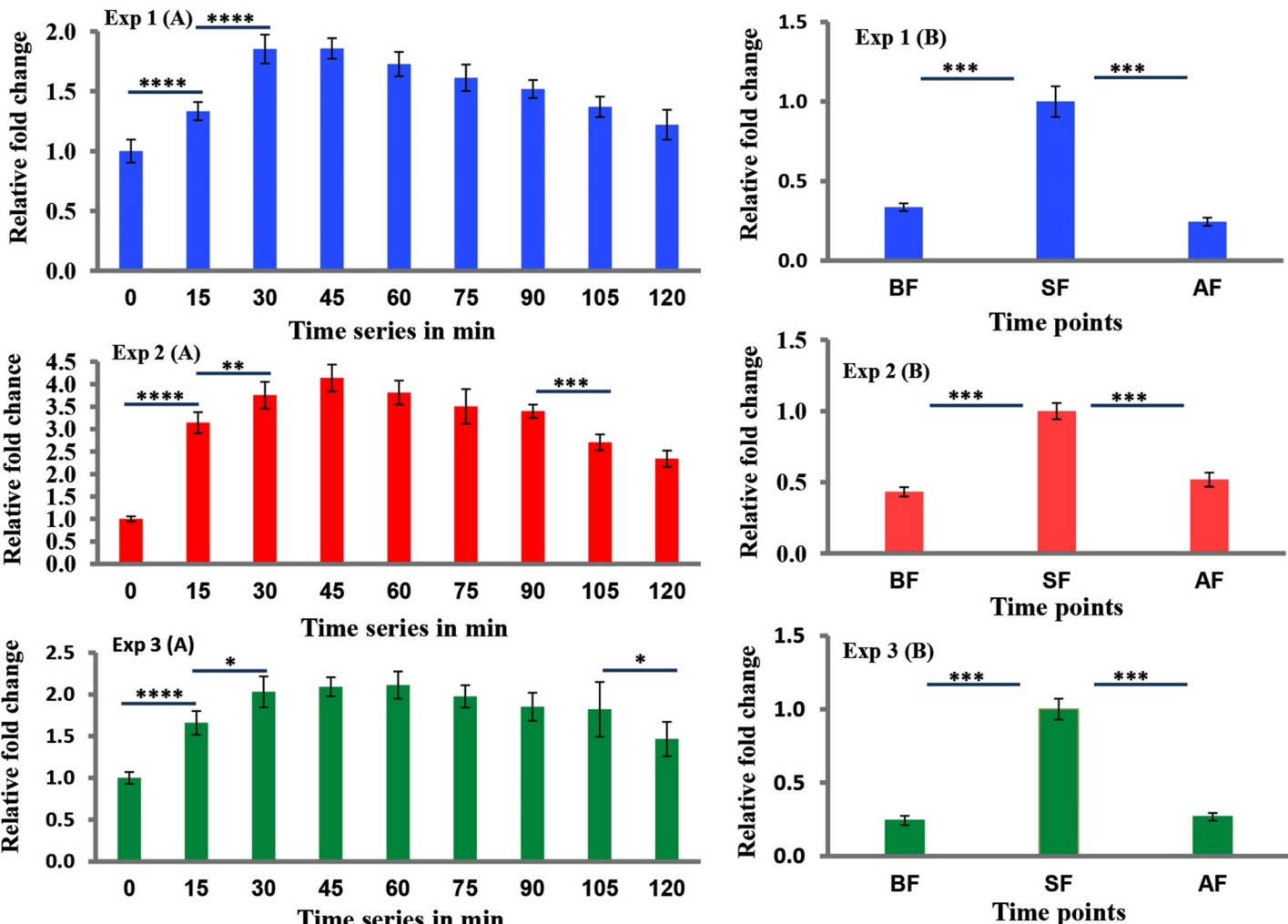

**Fig 1. Expression changes of the IEG *kakusei* during daily foraging of bees.** Data are shown as fold changes of *kakusei* expression level at different time points with respect to t0 (mean value was set to 1 at this time point). The blue, red and green bar graphs represent three independent experiments (experiment 1, experiment 2 and experiment 3 respectively) with each bar representing the *kakusei* expression level at each time point. Graphs at the left (Experiment series A) represent *kakusei* level from t0 (14.00 hrs) to t120 (16.00 hrs). Graphs at the right represent *kakusei* level at the start of foraging (SF/t0: 14.00 hrs) and before-foraging (BF: 09.00 hrs) or after foraging (AF: 18.00 hrs). Each time point has sample size of n = 5. For statistics one-way ANOVA with Turkey- Kramer post-hog multiple comparison test were performed and number of asterisk symbol represents * P < 0.05, ** P < 0.01, *** P < 0.001, and **** P < 0.0001.

## Testing for time-dependent food reward foraging effect on *kakusei* expression

To further examine whether the *kakusei* expression depends on the time of feeding, the sucrose solution was presented at 11.00 hrs, three hours ahead of the usual feeding time instead of 14.00 hrs (it may be recalled that the bees were fed every day at 14.00 hrs in the previous experiments). As before the marked bees were collected over a period of 1 hour, at 0min, 15min, 30min, 45min and 60min, as described in method section. A similar pattern of overexpression of *kakusei* was observed as in the previous experiment of food reward foraging in which the collection began at 14:00 hrs. The increase continued from 0min to 15min (P = <0.0001) and 15 min to 30min (P = 0.0195), and then showed no further increase ($P_{30m-45m}/P_{45m-60m}$ = not significant) (Fig 2D and S2 Table). This further reveals that *kakusei* role is independent of feeding time during the day, but responds to the reward of food during foraging.

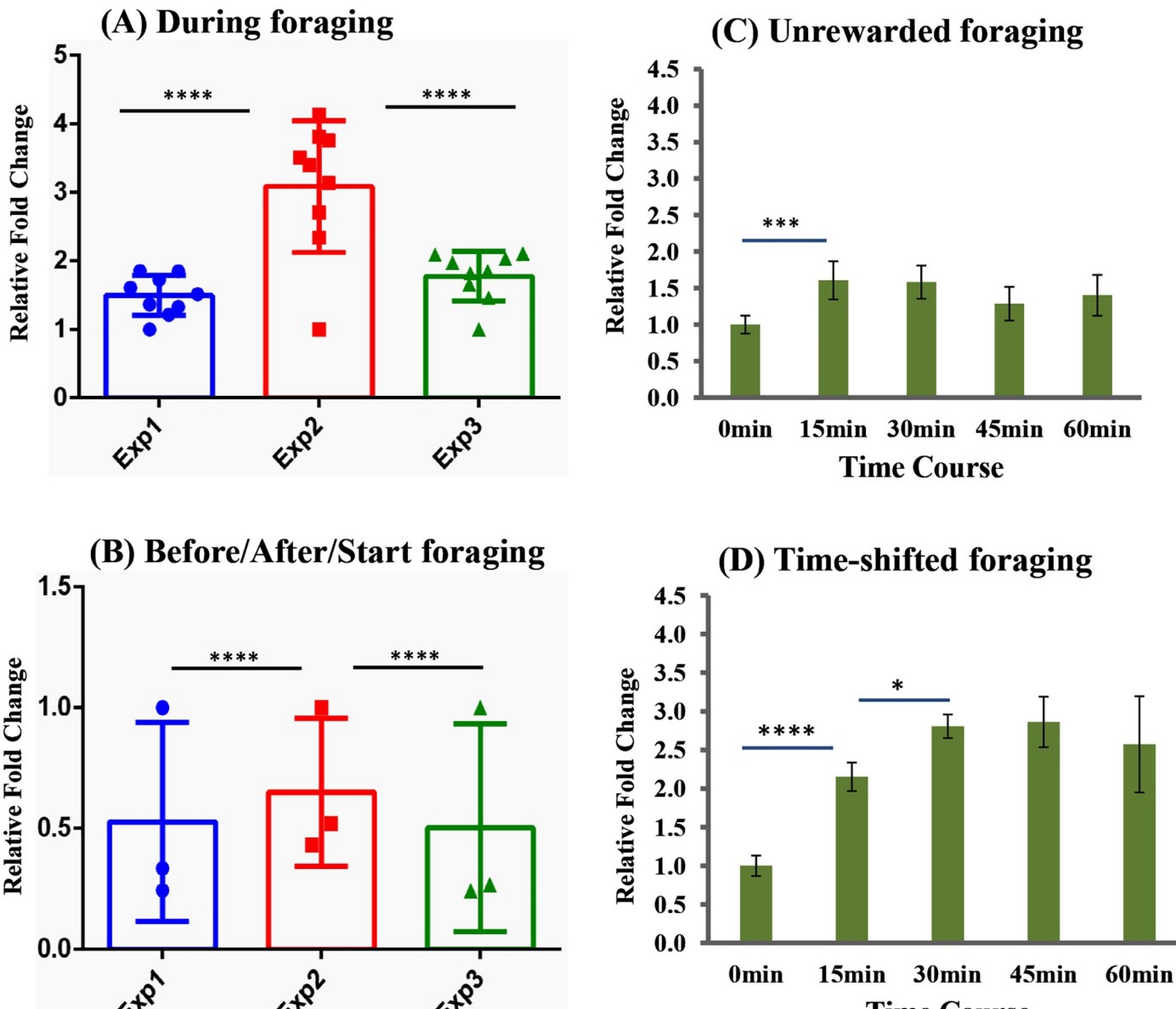

**Fig 2.** Comparison of three independent experiments during two hours of foraging (**A**) and before/after foraging (**B**), and expression changes of IEG *kakusei* with unrewarded (**C**) and different foraging time (**D**). **A&B**: The three independent experiments shown in Fig 1 were examined for interaction using two-way ANOVA. The bars represent mean *kakusei* expression in each experiment and the dots represent the mean level of *kakusei* at various time points. The blue, red and green color bars/dots represent experiment 1, experiment 2 and experiment 3 respectively. **C&D**: The *kakusei* expression profile is presented as fold changes with respect to t0 (mean value was set as 1 at this time point) and each green bar graphs represents *kakusei* levels at each time point. Fig 2C and 2D represent data for unrewarded and time-shifted foraging respectively. Each time point has sample size of n = 5. For statistics one-way ANOVA with Turkey- Kramer post-hog multiple comparison test were performed. The number of asterisk symbol represents * P < 0.05, ** P < 0.01, *** P < 0.001, and **** P < 0.0001.

## Discussion

The importance of using immediate early genes as tool for finding molecular and cellular mechanisms underlying neuronal network and behaviors in honey bees has been well emphasized in the recent studies [11,12,28,29]. Keeping this idea as central, this study extends our recent report [12] on the search for immediate early genes (IEGs) that could be used as

research tool for finding the molecular and cellular mechanisms underlying social behavior using foraging of honey bees as model system. As the foraging of bees consists of systematically well organized social behaviors that include learning, memory, social interaction and communication etc; honey bee foraging has been extensively studied in various research objectives in connection to social behavior [2,5,6]. Noting that honey bees learn food sources, identify the food location, memorize the place, interact and communicate among the foragers, as can be clearly observed during the time of their daily foraging; it is a promising approach to find the genes whose expression is increased upon foraging and to examine their roles in these behavioral routines. In the present study the possible function of the newly discovered insect IEG *kakusei* is presented, suggesting its role during the foraging of bees. A possible function of the gene in learning of food reward during foraging had also been observed.

*kakusei* is an insect immediate early gene recently identified by Kubo and his group in the seizure induced bees by cold or $CO_2$ treatment, and is transcribed into a non-coding neuronal RNA [20]. They found that the transcript was prominently increased in the small-type Kenyon cells of the dancers and foraging bee brains, suggesting involvement of the gene in foraging behavior. However, its involvement during the course of foraging had never been studied before. This is the first report on transient overexpression of *kakusei* during foraging. We have further examined the possible role of the gene before and after foraging. After food reward *kakusei* was immediately overexpressed over a short period of time from 15 to ca. 45 to 60 min, and then started decreasing. Interestingly, without food reward, there was a short time overexpression at 15min and no further increase of the gene level. This indicates that food reward is essential for the increase and sustaining of higher *kakusei* levels during foraging, and suggests a possible role of *kakusei* in learning and memorizing of food reward during foraging. The possibility of *kakusei* involvement in learning and memory was previously indicated in the earlier report by Kiya et al., 2007 [20] as they reported overrepresentation of *kakusei* among the re-orienting Kenyon cells of bees assumed to be foragers. This is in line with the notion that re-orientation flight was the mechanism of flight behavior that was practiced by those bees for memorizing the hive location, thus indicating a role of *kakusei* in the information integration during foraging flight, which is an essential part for dance communication [20]. We suspect that the *kakusei* gene might also be involved in social interaction and communication among the foraging bees, linking through different molecular pathways as these behaviors were clearly observed among the foragers over the two hours of foraging. Future research on this direction will be able to give a valid comment with evidence on it. Moreover, the suggestive evidence of the present study on the role of *kakusei* in learning needs to be rigorously tested further, using other experiments such as proboscis extension reflex (PER), in order to draw a conclusive answer which will be carried out in our further research. Our recent study by Singh et al., 2018 [12] had demonstrated involvement of two IEGs, *egr-1*and *hr38*, and downstream genes such as *ecr*, *dopecr*, *ddc*, and *dopr2* during daily foraging. Moreover, the role of *egr-1* and *hr38* in learning of food reward during foraging and memory processing was also suggested. Therefore, coordination of *kakusei* and the two IEGs *egr-1* and *hr38* during foraging behavior and learning and memory of food reward during foraging is highly assumable that would be tested in our future works. Since *kakusei* is a non coding gene, this study also clearly supports the dynamic role of non-coding genes in daily routine behavior.

On the other hand, earlier studies on songbird (Zebra finch) showed profound involvement of egr-1 gene in learning and memory [30]. In the *egr-1* transgenic cricket (*Gryllus bimaculatus*), behaviorally relevant neural circuits was also visualized at cellular resolution [31]. In rodent system, several studies had revealed *Egr-1* roles in synaptic plasticity and memory formation in a variety of memory systems including amygdala-dependent memory consolidation processes [32]. Moreover, the functions of *hr38* in neural activity in *Bombyx mori* (silkmoth)

and *Drosophila melanogaster* (fruit fly) and in honey bees were also revealed [14,33]. It is highly promising to do further research with *egr-1*, *hr38* and *kakusei* using honeybees as model system for further understanding on learning and memory and the finding on honeybees will be useful across animal kingdom.

## Conclusion

Previous reports have already indicated involvement of immediate early genes could be used as neural marker and their association in social behaviors such as learning and memory or memory processing. Subsequently, the importance of using immediate early genes as tool for finding molecular and cellular basis of these behaviors in honey bees had been thoroughly discussed [11,12,28,29]. Our recent report by Singh et al., has provided further evidence that the two IEGs *egr-1* and *hr38* are immediately and transiently expressed during honey bee foraging and further that they are involved in learning and memory processing [12]. The present finding on the recently discovered IEG *kakusei* is an addition to the circle, thus increasing the number of IEGs in elucidating molecular and cellular signaling underlying social behaviors, using honey bee foraging as model system. In addition to the available reports from different studies, further studies on how the three IEGs *egr-1*, *hr38* and *kakusei* coordinate in completing the foraging task via their specific role in the learning, memory and their roles in communication and interaction could contribute in mapping molecular and cellular pathways to these behavior. Thereby the mechanisms of social behavior in honey bees could be opened up in more details and mapped which could be further translated to cognitively complex animals and even to human [34].

## Supporting information

**S1 Fig.** Gene expression profile for c-Jun **(A)**, Dop1 **(B)**, GluR **(C)**, Erk7 **(D)** and 5-HT2α during the daily foraging of honey bees.
(DOCX)

**S1 Table. Summarized result for three replicate experiments of *kakusei*.**
(DOCX)

**S2 Table. Summarized result for time trained feeding effect and unrewarded foraging on *kakusei* expression.**
(DOCX)

## Acknowledgments

Dr. Ned Mantei, Department of Biology, Molecular Health Sciences, ETH Zurich, carefully and thoroughly read the manuscript. He provided valuable suggestions and advices and immensely helped in editing the manuscript. It is great honor to thank Dr. Ned Mantei for his kind and wholehearted support in making the manuscript to this form.

Dr. Axel Brockmann is gratefully acknowledged for providing the facilities in his lab to work and complete this project. Dr. Brockmann also read the manuscript and provided valuable comments. Ms. Neha Tanwar and Dr. Sophia Liyang kindly supported during experimental sample preparation.

## Author Contributions

**Conceptualization:** Asem Surindro Singh, Machathoibi Chanu Takhellambam.

**Data curation:** Asem Surindro Singh, Machathoibi Chanu Takhellambam.

**Formal analysis:** Asem Surindro Singh.

**Funding acquisition:** Asem Surindro Singh.

**Investigation:** Asem Surindro Singh, Machathoibi Chanu Takhellambam.

**Methodology:** Asem Surindro Singh.

**Project administration:** Asem Surindro Singh.

**Resources:** Asem Surindro Singh, Machathoibi Chanu Takhellambam.

**Software:** Asem Surindro Singh.

**Supervision:** Asem Surindro Singh.

**Validation:** Asem Surindro Singh, Machathoibi Chanu Takhellambam.

**Visualization:** Asem Surindro Singh, Machathoibi Chanu Takhellambam.

**Writing – original draft:** Asem Surindro Singh.

**Writing – review & editing:** Asem Surindro Singh, Machathoibi Chanu Takhellambam, Pamela Cappelletti, Marco Feligioni.

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
