## [Decision Letter · Decision Letter 0]

14 Oct 2019

PONE-D-19-23918

Immediate early gene kakusei plays a role in the daily foraging and learning of honey bees

PLOS ONE

Dear Dr Singh,

Thank you for submitting your manuscript to PLOS ONE. After careful consideration, we feel that it has merit but does not fully meet PLOS ONE’s publication criteria as it currently stands. Therefore, we invite you to submit a revised version of the manuscript that addresses the points raised during the review process.

Although Reviewer 1 recommended rejection, I feel that you could address these concerns, particularly the ones about overstating your curent findings (see comments of Reviewer 2 as well) and better acknowledging and placing your work in the context of prior work with this gene and similar genes (see Reviewer 1 comments). Both reviewers will be invited to re-review your submission.

We would appreciate receiving your revised manuscript by Nov 28 2019 11:59PM. To enhance the reproducibility of your results, we recommend that if applicable you deposit your laboratory protocols in protocols.io, where a protocol can be assigned its own identifier (DOI) such that it can be cited independently in the future. For instructions see: http://journals.plos.org/plosone/s/submission-guidelines#loc-laboratory-protocols

We look forward to receiving your revised manuscript.

Kind regards,

James C. Nieh, Ph.D.

Academic Editor

PLOS ONE

**Journal Requirements:**

2. Our internal editors have looked over your manuscript and determined that it may be within the scope of our Neuroscience of Reward and Decision Making Call for Papers. This collection of papers is headed by a team of Guest Editors for PLOS ONE: Stephanie Groman, Satoshi Ikemoto, Jane Taylor and Robert Whelan. With this Collection we hope to bring together researchers working on a wide range of disciplines, from animal subjects research, computational approaches and patient-centered research. Additional information can be found on our announcement page: https://collections.plos.org/s/reward-and-decision-making. If you would like your manuscript to be considered for this collection, please let us know in your cover letter and we will ensure that your paper is treated as if you were responding to this call. Agreeing to be part of the call-for-papers will not affect the date your manuscript is published. If you would prefer to remove your manuscript from collection consideration, please specify this in the cover letter.

3. Please include your tables as part of your main manuscript and remove the individual files. Please note that supplementary tables (should remain/ be uploaded )as separate "supporting information" files

4. Thank you for stating that “The funders had no role in study design, data collection and analysis, decision to publish, or preparation of the manuscript” in your financial disclosure.

Please also provide the name of the funders of this study (as well as grant numbers if available) in your financial disclosure statement.

5.  Please amend your Data availability statement to indicate exactly where the data can be found, and how other researchers can go about obtaining the dataset. Please include the relevant URLs, DOIs, or accession numbers within your revised cover letter. For a list of acceptable repositories, please see http://journals.plos.org/plosone/s/data-availability#loc-recommended-repositories. Any potentially identifying patient information must be fully anonymized.

Please amend your Data availability statement to indicate where the data can be found, and how other researchers can go about obtaining the dataset.

**Comments to the Author**

1. Is the manuscript technically sound, and do the data support the conclusions?

Reviewer #1: Partly

Reviewer #2: Yes

2. Has the statistical analysis been performed appropriately and rigorously? 

Reviewer #1: Yes

Reviewer #2: Yes

3. Have the authors made all data underlying the findings in their manuscript fully available?

Reviewer #1: Yes

Reviewer #2: No

4. Is the manuscript presented in an intelligible fashion and written in standard English?

Reviewer #1: Yes

Reviewer #2: No

5. Review Comments to the Author

Reviewer #1: This study analyzed the expression time-course of immediate early gene (IEG) kakusei, which encodes a non-coding RNA, during repetitive foraging flights of honeybee workers. The analysis of neural IEGs including kakusei would give insight into molecular mechanisms underlying honeybee foraging. However, I think that the present study contains some serious problems.

As the authors mentioned, the experimental design in this study is totally the same as it described in their previous study, which analyzed the other IEGs, egr-1 and Hr38. Although the authors assert that “the expansion of previous study”, I think that this is only “reuse”. Furthermore, in conlusion section, the authors note that “The present finding on recently discovered IEG kakusei is an addition into the circle, thus increases the number of IEGs and ultimately more IEG choices available that could be used as search tool in finding molecular and cellular signaling underlying social behaviors, using honey bee foraging as model system.” However, some previous studies (e.g., Kiya et al. 2011, Ugajin et al. 2018, and Sommerlandt et al. 2018) have already discussed this point. It is difficult for me to find what this study newly provides to the field of behavioral and/or molecular biology of honeybees.

I am also afraid that this manuscript contains a lot of overstating.

For example,

• Title: kakusei plays a role in ~

• Abstract: this study describes a fundamental role of the IEG kakusei ~

• Discussion: unraveling its role~, further examined the function of ~

This study does not perform any functional analysis of kakusei, but only examines the correlation between kakusei expression and time-course of foraging flight or food reward. The authors should entirely adjust the tone of their argument.

Moreover, in discussion section, the authors mention as if kakusei involves in the process of associative learning (“this finding of kakusei involvement in learning and memory”). Shorter span upregulation of kakusei during the unrewarded foraging experiment only indicates the relationship between sustained kakusei expression and food reward. There are other possible factors resulting in the low level induction of kakusei during the unrewarded foraging, such as a lack of dance behavior, and lower motivation to fly toward the empty feeder. Further experiment (e.g. PER experiment) is needed to examine an involvement of IEG expression in associative learning.

Minor points:

The results of statistical analysis described in TableS1 are not correctly reflected in Figure1.

Although text says that there is no significant difference among three independent trials of before/after/start foraging experiment, Figure2B still have asterisk symbols.

Gene name should be italicized (Title!).

Von Frisch - von Frisch

What does “09.00 hrs later” mean? 9 hours later? Or 9:00PM? Sampling time and procedure is very important for this study.

Reviewer #2: Review for: Immediate early gene kakusei plays a role in the daily foraging and learning of honey bees

This paper is about a non-coding gene, kakusei, that is found in the honey bee brain that seems to play a role in foraging and receiving reward. Generally, the authors collect bees after they have foraged for the day, before they have started foraging for the day, at different time points during their foraging day, at different time points during a time-shifted foraging day, and as they forage but get no reward. The authors find that this gene is upregulated as bees locate food, and if food continues to be found, but is quickly downregulated when there is no food, and is downregulated slowly if food is provided over time. It does not seem to be associated with time-shifted foraging. Overall, I believe the authors present very interesting results, and illustrate a relatively new area of behavioral genetics that deserves attention. However, the manuscript is confusing at times. Further, the authors overstate their results, as they do not test whether kakusei is related to associative learning, just with food reward during foraging. I have made suggestions for changes can be implemented to help the reader (especially readers who are not experts in this field) follow along.

Introduction

Line numbers would be helpful for review.

The introduction beginning with such a strong emphasis on waggle dance lead me to think it was going to be about waggling – although the authors do not examine the waggle dance in relation to kakusei regulation. The authors can give an overview of foraging behavior with short background of the waggle dance, but not such a large portion of the first paragraph.

I suggest breaking up the third paragraph. It has a lot of important information – especially for non-molecular biologists - but is long. A natural break could be right before “In our recent work…”

From the MS: “The IEG c- jun (also known as jun-related antigen, jra in fruit fly) and egr-1 have been used as neuronal markers for the identification in honey bees of specific brain regions involved in the time memory as well as in, innate and learned behaviours [17,18,19].”

This sentence seems a little grammatically off to me – I think the comma before innate is unnecessary

Methods

Methods are clear and concise.

What brand and type of paint markers was used to mark bees?

How many colonies were involved in the experiment? Are they represented across all experiments and samples?

Statistical analysis are effective for these data.

Controls with other IEGs and orthologs are effective.

Single-cohort age-matched bees would be best to use in experiments like these.

Results

Please report statistical analysis performed with stated p-value.

From the MS: “This result demonstrates that Kakuei is involved in associative learning.”

You have not shown that this gene is upregulated in associative learning – another test of associative learning, such as PER, needs to be performed before that conclusion can be made. Kakusei seems to be involved with reward during foraging, but not necessarily with learning. Also Kakusei is spelled incorrectly in that sentence.

Discussion

The link to associative learning has not been established with this study, and I suggest reducing the confidence of this language.

I agree that you have shown this, quoted from the MS: “This indicates that food reward is essential for the increase and sustaining of higher kakusei levels during foraging,…”

I do not agree that you have shown this: “…thus underlying the role of kakusei in associative learning during foraging.”

You have not shown that honey bees are in fact learning during your assay.

One experiment I would love to see to disentangle whether kakusei is upregulated when bees receive a reward or have learned is by doing PER on age matched foragers in the lab that haven’t foraged – and maybe to even see if it’s associated with strength of reward. It is a very easy test and would allow you to make the conclusions about learning.

From the MS: Thereby the hidden mechanisms of social behavior in honey bees could be opened up and mapped which could be further translated to higher animals and man [28].

Again, some of the language is unclear: Mechanisms are not hidden – aspects to gene function like non-coding regions are just now being discovered with relatively new techniques, which is really cool! Also, this hierarchical language of "higher animals" is not helpful in a scientific paper – “cognitively complex” would be a more accurate way to describe some animals (although bees are very cognitively complex). Also please use “humans” instead of “man”, although it is redundant after stating more cognitively complex animals.

Figures

Figures are effective and relatively clear.

In figure 2, I suggest using more descriptive X axis labels – I had to go back to the results to remind myself which experiment was which (although they are not clearly labeled experiment 1,2, and 3 there either). I suggest: During Foraging, Unrewarded Foraging, Time-Shifted Foraging. A and B labels are also missing.

For all figures, do the colors mean anything? I suggest to color-code by experiment, and keep it consistent though the two figures. Also explicitly state this in figure legends

6. PLOS authors have the option to publish the peer review history of their article (what does this mean?). If published, this will include your full peer review and any attached files.

Reviewer #1: No

Reviewer #2: No

---

## [Author Response · Author response to Decision Letter 0]

28 Jan 2020

We are very grateful to the reviewers for the great effort in thorough reading of the manuscript and for providing valuable comments and suggestions that help to increase the quality of the manuscript. 

The answers to the comments of the reviewers are given in the following:

Reviewer #1: This study analyzed the expression time-course of immediate early gene (IEG) kakusei, which encodes a non-coding RNA, during repetitive foraging flights of honeybee workers. The analysis of neural IEGs including kakusei would give insight into molecular mechanisms underlying honeybee foraging. However, I think that the present study contains some serious problems. As the authors mentioned, the experimental design in this study is totally the same as it described in their previous study, which analyzed the other IEGs, egr-1 and Hr38. Although the authors assert that “the expansion of previous study”, I think that this is only “reuse”.

Ans: The reviewer has pointed out very correctly. It is true that our experimental design in this study is same with that of the previous study. The samples that we used are also same; so, to say “reuse” is a correct word at this point. On the other hand, investigating kakusei and the other 5 new genes which we have not studied before to the same experimental design set up is what we mean to “expansion to the previous study”. To enclose reviewer’s comment we have inserted “the same samples have been reused in this study, in the method section line no. 107. 

Thus we have also mentioned in discussion section a possible connection with kakusei and the two genes egr-1 and hr38 of the previous report.

Furthermore, in conlusion section, the authors note that “The present finding on recently discovered IEG kakusei is an addition into the circle, thus increases the number of IEGs and ultimately more IEG choices available that could be used as search tool in finding molecular and cellular signaling underlying social behaviors, using honey bee foraging as model system.” However, some previous studies (e.g., Kiya et al. 2011, Ugajin et al. 2018, and Sommerlandt et al. 2018) have already discussed this point. It is difficult for me to find what this study newly provides to the field of behavioral and/or molecular biology of honeybees. 

Ans: We have adjusted the tone adding the first line in discussion section citing 4 references, and also modification is made in the conclusion section. Moreover we have cited more reference at the last part of the discussion section.

I am also afraid that this manuscript contains a lot of overstating. 

For example, 

• Title: kakusei plays a role in ~ 

Ans: We have added potentially to neutralize the overstatement, in the title. 

• Abstract: this study describes a fundamental role of the IEG kakusei ~ 

Ans: We have substituted the word “fundamental” to “possible”, line no. 17.

• Discussion: unraveling its role~, further examined the function of ~ This study does not perform any functional analysis of kakusei, but only examines the correlation between kakusei expression and time-course of foraging flight or food reward. The authors should entirely adjust the tone of their argument. 

Ans: We have also adjusted this tone by replacing “unraveling” with “suggesting” line no. 260, and the “function” by “possible role” line no. 267. 

Moreover, in discussion section, the authors mention as if kakusei involves in the process of associative learning (“this finding of kakusei involvement in learning and memory”). Shorter span upregulation of kakusei during the unrewarded foraging experiment only indicates the relationship between sustained kakusei expression and food reward. There are other possible factors resulting in the low level induction of kakusei during the unrewarded foraging, such as a lack of dance behavior, and lower motivation to fly toward the empty feeder. Further experiment (e.g. PER experiment) is needed to examine an involvement of IEG expression in associative learning. 

Ans: We have removed “associative” word from the entire text of the manuscript. We are very interested to do PER experiment. However, as all the authors currently work at different places on different projects, we are not able to predict when the facilities will be available to do this experiment. But this experiment is one of our priorities in the future plan. 

Minor points: 

The results of statistical analysis described in TableS1 are not correctly reflected in Figure1.

Ans: We have corrected in the figure. 

Although text says that there is no significant difference among three independent trials of before/after/start foraging experiment, Figure2B still have asterisk symbols. 

Ans: We have corrected. The result of the Fig 1 analysis was also mistakenly re-written there. What we wanted to write was the similarity between Fig 1 and Fig 2. We have reconstructed the sentence for more clarity.

Gene name should be italicized (Title!). 

Ans: We have corrected.

Von Frisch - von Frisch 

Ans: We have corrected.

What does “09.00 hrs later” mean? 9 hours later? Or 9:00PM? Sampling time and procedure is very important for this study. 

Ans: We have changed to 18.00 hours, line no. 124. 

Reviewer #2: Review for: Immediate early gene kakusei plays a role in the daily foraging and learning of honey bees 

This paper is about a non-coding gene, kakusei, that is found in the honey bee brain that seems to play a role in foraging and receiving reward. Generally, the authors collect bees after they have foraged for the day, before they have started foraging for the day, at different time points during their foraging day, at different time points during a time-shifted foraging day, and as they forage but get no reward. The authors find that this gene is upregulated as bees locate food, and if food continues to be found, but is quickly downregulated when there is no food, and is downregulated slowly if food is provided over time. It does not seem to be associated with time-shifted foraging. Overall, I believe the authors present very interesting results, and illustrate a relatively new area of behavioral genetics that deserves attention. However, the manuscript is confusing at times. Further, the authors overstate their results, as they do not test whether kakusei is related to associative learning, just with food reward during foraging. I have made suggestions for changes can be implemented to help the reader (especially readers who are not experts in this field) follow along. 

Ans: We appreciate for the kind words. We have removed “associative” word throughout the text of the manuscript.

Introduction 

Line numbers would be helpful for review. 

Ans: We have added line numbers this time.

The introduction beginning with such a strong emphasis on waggle dance lead me to think it was going to be about waggling – although the authors do not examine the waggle dance in relation to kakusei regulation. The authors can give an overview of foraging behavior with short background of the waggle dance, but not such a large portion of the first paragraph.

Ans: We have reduced the length of the sentence to about half from the previous one, line no. 35-36.

I suggest breaking up the third paragraph. It has a lot of important information – especially for non-molecular biologists but is long. A natural break could be right before “In our recent work…” 

Ans: We have broken the paragraph right before “in our recent work” line no. line 65.

From the MS: “The IEG c- jun (also known as jun-related antigen, jra in fruit fly) and egr-1 have been used as neuronal markers for the identification in honey bees of specific brain regions involved in the time memory as well as in, innate and learned behaviours [17,18,19].” 

This sentence seems a little grammatically off to me – I think the comma before innate is unnecessary.

Ans: We have removed the coma accordingly, line no. 75. 

Methods Methods are clear and concise.

Ans: We appreciate for the complement. 

What brand and type of paint markers was used to mark bees?

Ans: It is Uni POSCA Paint Markers (Uni Mitsubishi Pencil, UK). We have mentioned this also in the method section, line no. 118.

How many colonies were involved in the experiment? Are they represented across all experiments and samples? 

Ans: We have use two colonies. The colony 1 was used for experiment 1 and 2, and colony 2 was used for experiment 3. For further clarity in the manuscript, we have also inserted this in the result section, line no. 177-179. 

Statistical analysis are effective for these data.

 Ans: We appreciate for the complement.

Controls with other IEGs and orthologs are effective. 

Ans: We have done for c-jun another IEG but we found no difference. It is shown in supplementary figure. In our future work we will take it seriously to add more ortholog genes. 

Single-cohort age-matched bees would be best to use in experiments like these. 

Ans: We very much agree to it. It may be done by marking the bees from the larval stage. In our future experiment we will prioritize to it.

Results

Please report statistical analysis performed with stated p-value.

Ans: More statistics P-values are reported in the text, line no. 193, 228, 231, 243.

From the MS: “This result demonstrates that Kakuei is involved in associative learning.” 

You have not shown that this gene is upregulated in associative learning – another test of associative learning, such as PER, needs to be performed before that conclusion can be made. Kakusei seems to be involved with reward during foraging, but not necessarily with learning. Also Kakusei is spelled incorrectly in that sentence.

Ans: We have removed “associative” word from the entire text of the manuscript. We are very interested to do PER experiment. However, as all the authors currently work at different places on different projects, therefore we are not able to predict when the facilities will be available to do this experiment. But this experiment is one of our priorities in the future plan. kakusei spelling is also corrected.

Discussion 

The link to associative learning has not been established with this study, and I suggest reducing the confidence of this language. 

Ans: We have removed “associative” word from the entire text of the manuscript. 

I agree that you have shown this, quoted from the MS: “This indicates that food reward is essential for the increase and sustaining of higher kakusei levels during foraging,…” 

I do not agree that you have shown this: “…thus underlying the role of kakusei in associative learning during foraging.” You have not shown that honey bees are in fact learning during your assay. 

Ans: We have removed “associative” word from the entire text of the manuscript. 

The feeder plate usually contain feeder, but bees still arrived and landed on the feeder plate even when the empty plate was placed. We paint marked the bees at the first arrival, and they still continue to come to the feeder though fewer and fewer, they might be expecting they would be rewarded, thus we were able to collect the samples until 60min, without rewarding food. Since, we observed significantly increased kakusei level from t0 to t15, but reduced after 15min as there was continuous unrewarded of food, we assumed that the bees were motivated to food reward which they experienced previously. If this explanation is still unsatisfactory, we will consider removing learning also from the entire manuscript. 

One experiment I would love to see to disentangle whether kakusei is upregulated when bees receive a reward or have learned is by doing PER on age matched foragers in the lab that haven’t foraged – and maybe to even see if it’s associated with strength of reward. It is a very easy test and would allow you to make the conclusions about learning.

 Ans: We have removed “associative” word from the entire manuscript. We are very interested to do PER experiment. However, as all the authors currently work at different places on different projects, therefore we are not able to predict when the facilities will be available to do this experiment. But this experiment is one of our priorities in the future plan. kakusei spelling is also corrected.

From the MS: Thereby the hidden mechanisms of social behavior in honey bees could be opened up and mapped which could be further translated to higher animals and man [28]. 

Again, some of the language is unclear: Mechanisms are not hidden – aspects to gene function like non-coding regions are just now being discovered with relatively new techniques, which is really cool! 

Ans: We have removed “hidden”. We appreciate for the complement.

Also, this hierarchical language of "higher animals" is not helpful in a scientific paper – “cognitively complex” would be a more accurate way to describe some animals (although bees are very cognitively complex). Also please use “humans” instead of “man”, although it is redundant after stating more cognitively complex animals. 

Ans: We have followed the good advice and changed accordingly, line no. 307.

Figures 

Figures are effective and relatively clear. 

Ans: We appreciate for the complement.

In figure 2, I suggest using more descriptive X axis labels – I had to go back to the results to remind myself which experiment was which (although they are not clearly labeled experiment 1,2, and 3 there either). I suggest: During Foraging, Unrewarded Foraging, Time-Shifted Foraging. A and B labels are also missing. 

Ans: We have made it clearer in the legends to the figure and also we have changed the color pattern for easier viewing. We have also labeled experiment 1, 2 and 3 more clearly on the figure and the same is reflected on the results. We have also labeled A and B in Fig 2 which was missing earlier. 

For all figures, do the colors mean anything? I suggest to color-code by experiment, and keep it consistent though the two figures. Also explicitly state this in figure legends

Ans: We have changed the color pattern in the Fig 1, three different colors for three experiments, same color for both A and B for each experiment. We have also mentioned color representations in the figure legends.

---

## [Decision Letter · Decision Letter 1]

19 Feb 2020

PONE-D-19-23918R1

Immediate early gene kakusei potentially plays a role in the daily foraging and learning of honey bees

PLOS ONE

Dear Dr Singh,

Thank you for submitting your manuscript to PLOS ONE. After careful consideration, we feel that it has merit but does not fully meet PLOS ONE’s publication criteria as it currently stands. Therefore, we invite you to submit a revised version of the manuscript that addresses the points raised during the review process.

Please address the comments of both reviewers, particularly the recommendation of reviewer 1 about linking kakusei expression to learning.

We would appreciate receiving your revised manuscript by Apr 04 2020 11:59PM. To enhance the reproducibility of your results, we recommend that if applicable you deposit your laboratory protocols in protocols.io, where a protocol can be assigned its own identifier (DOI) such that it can be cited independently in the future. For instructions see: http://journals.plos.org/plosone/s/submission-guidelines#loc-laboratory-protocols

We look forward to receiving your revised manuscript.

Kind regards,

James C. Nieh, Ph.D.

Academic Editor

PLOS ONE

Reviewers' comments:

Reviewer's Responses to Questions

**Comments to the Author**

1. If the authors have adequately addressed your comments raised in a previous round of review and you feel that this manuscript is now acceptable for publication, you may indicate that here to bypass the “Comments to the Author” section, enter your conflict of interest statement in the “Confidential to Editor” section, and submit your "Accept" recommendation.

Reviewer #1: (No Response)

Reviewer #2: All comments have been addressed

2. Is the manuscript technically sound, and do the data support the conclusions?

Reviewer #1: Yes

Reviewer #2: No

3. Has the statistical analysis been performed appropriately and rigorously? 

Reviewer #1: Yes

Reviewer #2: Yes

4. Have the authors made all data underlying the findings in their manuscript fully available?

Reviewer #1: Yes

Reviewer #2: Yes

5. Is the manuscript presented in an intelligible fashion and written in standard English?

Reviewer #1: Yes

Reviewer #2: Yes

6. Review Comments to the Author

Reviewer #1: The authors generally addressed my comments except for PER experiment. But they carefully adjusted the tone related to the role of kakusei for learning and memory throughout the revised manuscript. Unfortunately, there are still inconsistency between Figure1 and TableS1. For example, regarding experiment 1A, are there significant differences in kakusei expression only “t0 vs t15” and “t15 vs t30”? Please correct Figure1 precisely.

Reviewer #2: The explanation of learning is still not satisfactory. The authors have not fully tested the hypothesis that expression of kakusei is associated with learning. Changes in expression seen in this article could be due to a number of things – such as sensory reception of food or floral cues, digestion, navigation, etc. – which is still interesting! BUT much more work has to be done to link kakusei expression to learning. It is fine to propose in the intro and discussion, but it’s not explicitly tested here. I highly recommend removing any language about learning until it is explicitly tested.

The figures also seem to be low resolution – some of the letters and shapes are difficult to read.

7. PLOS authors have the option to publish the peer review history of their article (what does this mean?). If published, this will include your full peer review and any attached files.

Reviewer #1: No

Reviewer #2: No

---

## [Author Response · Author response to Decision Letter 1]

7 Mar 2020

Reviewer #1: The authors generally addressed my comments except for PER experiment. But they carefully adjusted the tone related to the role of kakusei for learning and memory throughout the revised manuscript. Unfortunately, there are still inconsistency between Figure1 and TableS1. For example, regarding experiment 1A, are there significant differences in kakusei expression only “t0 vs t15” and “t15 vs t30”? Please correct Figure1 precisely.

Ans: PER has been addressed in line no. 284-285 in this revision. Regarding the placing of asterisk symbols for all the significant differences in the figure 1, we have considered only the adjacent time points such as t0 vs t15” and “t15 vs t30, as noted by the reviewer; that is, we have not shown all the significant differences on the figure. This is for the reason that we wanted to show a clean figure as per the need of the description in the result and for further detail presentation of all the significant differences, Table S1 has been created. Nonetheless, we have also tried following the suggestion of the reviewer. However, we have kept the figure without changing the earlier form, because of the congestion of the lines, as shown below 

Reviewer #2: The explanation of learning is still not satisfactory. The authors have not fully tested the hypothesis that expression of kakusei is associated with learning. Changes in expression seen in this article could be due to a number of things – such as sensory reception of food or floral cues, digestion, navigation, etc. – which is still interesting! BUT much more work has to be done to link kakusei expression to learning. It is fine to propose in the intro and discussion, but it’s not explicitly tested here. I highly recommend removing any language about learning until it is explicitly tested. The figures also seem to be low resolution – some of the letters and shapes are difficult to read.

Ans: We have diluted about the learning in the discussion to a great extend and we have added our future work plans to give a conclusive comment on it. The changes are made at line nos., 272-274, 282-286 and 300-302. We are also very grateful to the editor’s advice in this regard. We would like to do further changes if it is needed. 

We have made the figures more clear this time.

---

## [Decision Letter · Decision Letter 2]

13 Mar 2020

PONE-D-19-23918R2

Immediate early gene kakusei potentially plays a role in the daily foraging and learning of honey bees

PLOS ONE

Dear Dr Singh,

Thank you for submitting your manuscript to PLOS ONE. After careful consideration, we feel that it has merit but does not fully meet PLOS ONE’s publication criteria as it currently stands. Therefore, we invite you to submit a revised version of the manuscript that addresses the points raised during the review process.

Please remove the words "and learning" from the manuscript title. 

We would appreciate receiving your revised manuscript by Apr 27 2020 11:59PM. To enhance the reproducibility of your results, we recommend that if applicable you deposit your laboratory protocols in protocols.io, where a protocol can be assigned its own identifier (DOI) such that it can be cited independently in the future. For instructions see: http://journals.plos.org/plosone/s/submission-guidelines#loc-laboratory-protocols

We look forward to receiving your revised manuscript.

Kind regards,

James C. Nieh, Ph.D.

Academic Editor

PLOS ONE

Reviewers' comments:

Reviewer's Responses to Questions

**Comments to the Author**

1. If the authors have adequately addressed your comments raised in a previous round of review and you feel that this manuscript is now acceptable for publication, you may indicate that here to bypass the “Comments to the Author” section, enter your conflict of interest statement in the “Confidential to Editor” section, and submit your "Accept" recommendation.

Reviewer #2: All comments have been addressed

2. Is the manuscript technically sound, and do the data support the conclusions?

Reviewer #2: Yes

3. Has the statistical analysis been performed appropriately and rigorously? 

Reviewer #2: Yes

4. Have the authors made all data underlying the findings in their manuscript fully available?

Reviewer #2: Yes

5. Is the manuscript presented in an intelligible fashion and written in standard English?

Reviewer #2: Yes

6. Review Comments to the Author

Reviewer #2: The title still contains the word "learning" which should be removed to reflect the changes in the manuscript.

7. PLOS authors have the option to publish the peer review history of their article (what does this mean?). If published, this will include your full peer review and any attached files.

Reviewer #2: No

---

## [Author Response · Author response to Decision Letter 2]

18 Mar 2020

Dear Editor, 

We are very grateful to you and to the reviewers for providing valuable comments and suggestions and advices that help to increase the quality of the manuscript for the publication in your reputed journal PLOS ONE. 

The answer to the comment of the reviewer is given as below:

Reviewer #2: The title still contains the word "learning" which should be removed to reflect the changes in the manuscript.

Edotor: Please remove the words "and learning" from the manuscript title

Ans: “and Learning” have been removed from the manuscript title.

---

## [Editor Report · Decision Letter 3]

20 Mar 2020

Immediate early gene kakusei potentially plays a role in the daily foraging of honey bees

PONE-D-19-23918R3

Dear Dr. Singh,

We are pleased to inform you that your manuscript has been judged scientifically suitable for publication and will be formally accepted for publication once it complies with all outstanding technical requirements.

With kind regards,

James C. Nieh, Ph.D.

Academic Editor

PLOS ONE
---

## [Editor Report · Acceptance letter]

9 Apr 2020

PONE-D-19-23918R3 

Immediate early gene *kakusei* potentially plays a role in the daily foraging of honey bees  

Dear Dr. Singh:

I am pleased to inform you that your manuscript has been deemed suitable for publication in PLOS ONE. Congratulations! Your manuscript is now with our production department. 

With kind regards,

on behalf of

Dr. James C. Nieh 

%CORR_ED_EDITOR_ROLE%

PLOS ONE